# CogMath: Evaluating LLMs' Authentic Mathematical Ability from a cognitive perspective

## Abstract

As large language models (LLMs) exhibit potential in solving complex mathematical tasks, increasing attention has been directed toward constructing benchmarks to evaluate their mathematical capabilities. However, existing benchmarks are either limited to specific task types (e.g., long-text problem understanding) or rely solely on a coarse measure of answer accuracy, making them insufficient for assessing a model's authentic mathematical proficiency. In this paper, we propose **CogMath**, which provides a comprehensive assessment of LLMs' mathematical abilities based on human cognitive processes. Specifically, inspired by cognitive theories, CogMath formalizes the reasoning process into 3 stages that align with human cognition: *problem comprehension*, *problem solving*, and *solution summarization*, and encompasses 9 fine-grained evaluation dimensions from perspectives such as numerical calculation, knowledge, and counterfactuals. In each dimension, to carry out a scientific evaluation, we develop an "*Inquiry-Judge-Reference*" multi-agent system, where the *Inquiry* agent generates inquiries that assess LLMs' mastery from this dimension, the *Judge* agent ensures the inquiry quality, and the *Reference* agent provides correct responses for comparison with the LLMs' actual performances. A LLM is considered to truly master a problem only when excelling in all inquiries from the 9 dimensions. In experiments, we evaluate 7 mainstream LLMs by applying CogMath to three benchmarks, which cover the full K-12 mathematical curriculum. The results reveal that the authentic mathematical capabilities of current LLMs are overestimated by 30-40%. Moreover, we locate their strengths and weaknesses across different stages/dimensions, offering constructive insights to further enhance their reasoning abilities.

## 1 Introduction

The rise of large language models (LLMs) has marked a pivotal moment in artificial intelligence. Particularly within the realm of mathematical reasoning, these models have made breakthroughs in solving complex mathematical problems. For example, GPT-4 has achieved over 75% accuracy on the high school competition-level MATH dataset (Hendrycks et al., 2021). More recently, the OpenAI-o1 model has surpassed 70% accuracy on the AIME math competition, placing it at a level comparable to the top 500 US high school students [1]. This remarkable progress has not only redefined the potential of AI in mathematics but also spurred a growing body of research dedicated to evaluating and understanding the mathematical proficiencies of these models.

To systematically assess the mathematical ability of LLMs, numerous benchmarks have been introduced. For instance, E-GSM (Xu et al., 2024) includes mathematical problems across four different length ranges to assess LLMs' generalization capabilities regarding problem text length. GSM-Plus (Li et al., 2024) introduces eight variants of GSM8K dataset (Cobbe et al., 2021) to investigate the robustness of LLMs. MPA (Zhu et al., 2024) rewrites four existing datasets based on five principles, confirming that the mathematical abilities of LLMs may be affected by data contamination. However, on one hand, these benchmarks tend to be overly task-specific, requiring the construction of particular types of mathematical problems (e.g., long-text problems) to investigate one or some specific aspects of a model's capabilities (e.g., long-text understanding). On the other hand, they rely on a coarse accuracy metric to evaluate the overall performance of models, without deeply assessing

---

[1] https://openai.com/index/openai-o1-mini-advancing-cost-efficient-reasoning/

their internal mathematical reasoning processes. Consequently, they are unable to fully grasp the entire spectrum of mathematical capabilities that LLMs possess.

In this paper, we propose **CogMath**, which offers a scientific and comprehensive evaluation of LLMs' mathematical abilities by delving into the cognitive stages of reasoning processes employed by humans. Specifically, psychological research points out that humans undergo three stages when reasoning about a mathematical problem (Schoenfeld, 2014; Lesh & Doerr, 2003; Dehaene et al., 1999): *problem comprehension*, *problem solving*, and *solution summarization*, which can be formalized as Eq (1). For a given problem $P$, humans first rely on a comprehension system $f_{comprehend}$ to grasp its semantics. Then, by combining the semantic information with the mathematical knowledge $K$, the logical solving system $f_{solve}$ in the human brain derives the answer. After obtaining the answer, humans organize the solving process to form a complete logical chain and summarize the solution into a coherent methodology, which is denoted as $f_{summarize}$.

$$Human\ Reasoning = f_{summarize} \circ f_{solve}(f_{comprehend}(P), K) \tag{1}$$

Corresponding to the three cognitive stages, we design nine evaluation dimensions to ensure a scientific and comprehensive assessment. Each dimension evaluates the LLM's performance in one stage from perspectives such as computation, knowledge, and counterfactual reasoning. For example, in *problem comprehension* stage, we assess the model's ability to handle different formulations of the same problem (e.g., paraphrasing or counterfactually removing conditions) to determine whether it truly understands the core meaning. In *problem solving* stage, we break down the solution into three orthogonal aspects: problem-solving strategy, numerical computation, and knowledge application, and evaluate LLMs in each aspect independently. In *Solution summarization* stage, we go beyond traditional forward evaluation by introducing intermediate step questions and backward reasoning tasks, testing whether the model can trace back through its reasoning pathway. Through these nine dimensions, we can systematically gain insights into both the strengths and weaknesses of LLMs.

Moreover, we design an "*Inquiry-Judge-Reference*" multi-agent system to carry out scientific evaluation in each dimension. The *Inquiry* agent is responsible for posing an inquiry about a problem from this dimension. These inquiries either 1) ask about the original problem text or the problem-solving steps, 2) rephrase the problem while maintaining the same difficulty and knowledge scope, or 3) construct "pseudo problems" to test the model's boundaries in counterfactual scenarios. To ensure the quality of inquiries, we design a *Judge* agent for each *Inquiry* agent to evaluate and refine its output. Besides, for each inquiry, we introduce a *Reference* agent to provide the correct answer, serving as a standard to evaluate whether a LLM's actual performance on that inquiry meets expectations. Compared with existing evaluations that rely solely on an answer accuracy, **CogMath considers a LLM to truly master a problem only after excelling in all inquiries in 9 dimensions**.

In experiments, we apply CogMath to the most representative mathematical benchmarks GSM8K and MATH, along with an additional dataset we collected, MExam, which is composed of real exam tests that cover the full K-12 curriculum. Then, we evaluate 7 mainstream LLMs including GPT-4 (Achiam et al., 2023), GPT-3.5-Turbo (OpenAI, 2023), Gemini-1.5-Flash (Team et al., 2023), Deepseek-v2.5 Liu et al. (2024a), Llama3-8B (Meta, 2024), Llama2-13B (Touvron et al., 2023), and Mixtral-8x7B-Instruct (MistralAITeam, 2023). Our key experimental findings are as follows [2]:

- The authentic mathematical capabilities of current LLMs are overestimated by 30-40%. For instance, GPT-4 has truly mastered only 39.7% and 67.1% of the problems in MATH and GSM8K datasets, respectively. Moreover, this overestimation is not solely attributable to data contamination, but rather to an excessive imitation of superficial patterns of reasoning.

- We locate the deficiency stage of LLMs. Weaker models (e.g., Llama2-13B) still struggle in *problem comprehension* stage, while stronger models (e.g., GPT-4, Deepseek-v2.5) face challenges primarily in *problem solving* stage, particularly in their mastery of knowledge.

- Confronted with a counterfactual setting, current LLMs may exhibit an inherent "overcorrection" behavior, automatically aligning with patterns from the training data.

- Existing prompting techniques, such as CoT and ICL, may fail to consistently and reliably improve the mathematical reasoning capabilities of LLMs.

---

[2]Code and data available at `https://anonymous.4open.science/r/CogMath-2743`.

## 2 RELATED WORK

### 2.1 LARGE LANGUAGE MODELS

Large language models (LLMs) have significantly advanced the field of natural language processing (NLP). Models like OpenAI-o1, GPT-4 (Achiam et al., 2023), and GPT-3.5-Turbo (OpenAI, 2023) have set new performance milestones across numerous NLP tasks, such as sentiment classification (Zhang et al., 2024b), question answering (Hendrycks et al., 2021), and translation (Wang et al., 2023a). To further enhance their reasoning and problem-solving abilities, several advanced techniques have been introduced. Among them, Chain-of-Thought (CoT) (Wei et al., 2022), Tree-of-Thought (ToT) (Yao et al., 2024), and Graph-of-Thought (GoT) (Besta et al., 2024) simulate structured and logical reasoning paths using chains, trees, and graphs, respectively, allowing models to handle multi-step problems more effectively. Program of Thought (PoT) (Chen et al., 2023a) and PAL (Gao et al., 2023) introduce formal programming and have the LLMs generate executable code, thereby performing more rigorous and deterministic computations. Another key development in LLMs is In-Context Learning (ICL) (Dong et al., 2022), where the model can learn from a few examples to generalize and solve unseen problems. In addition, there are many other key techniques, such as self-consistency (Wang et al., 2023b) and Retrieval-Augmented Generation (RAG) (Chen et al., 2024). We refer the readers to a more detailed survey conducted by Zhao et al. (2023).

### 2.2 EVALUATION ON LLMS' MATHEMATICAL ABILITY

We categorize existing mathematical benchmarks from two perspectives: problem difficulty and problem types. In terms of difficulty, MATH (Hendrycks et al., 2021) and CHAMP (Mao et al.) are representative of high school competition-level datasets, while GSM8K (Cobbe et al., 2021) and MAWPS (Koncel-Kedziorski et al., 2016) are composed of elementary-level math word problems. From the perspective of problem types, E-GSM (Xu et al., 2024) includes four categories of math problems of varying lengths to evaluate LLMs' generalization on longer contexts, TheoremQA (Chen et al., 2023b) and MathBench (Liu et al., 2024b) test LLMs' ability to prove and apply theorems, while MathVista (Lu et al., 2024) and GeoEval (Zhang et al., 2024a) focus on visual reasoning and deeper geometric reasoning problems. To mitigate the impact of data contamination, some studies introduce perturbations into existing benchmarks, such as GSM-Plus Li et al. (2024) and MPA (Zhu et al., 2024) that consist of eight/five variations of GSM8K, respectively. However, we argue that existing benchmarks are often task-specific, which rely on particular type of problems to test one or some specific capabilities (e.g., long-text understanding). Moreover, these benchmarks lack in-depth exploration of models' reasoning processes, instead relying on coarse overall accuracy metrics, which makes it difficult to precisely identify at which cognitive stage the LLM encounters issues. Consequently, this limits the ability to provide interpretable guidance for improving LLMs.

## 3 OUR EVALUATION FRAMEWORK: COGMATH

To achieve a comprehensive and scientific evaluation, we draw inspiration from how humans solve mathematical problems. Specifically, psychological theories indicate that human reasoning process consists of three stages: *problem comprehension*, *problem solving*, and *solution summarization* (Schoenfeld, 2014; Lesh & Doerr, 2003; Dehaene et al., 1999). As formalized in Eq. (1), these three stages build upon each other, with each stage taking the output of the previous one as input. *Problem comprehension* involves analyzing problem $P$'s information, such as word semantics, text structure, and given conditions. *Problem solving* stage combines the problem information with relevant knowledge $K$ (e.g., the concept of word "half", area formula of rectangle) to infer a solution. Finally, in *solution summarization* stage, humans engage in self-summarization, reviewing their thought processes, organizing clear logical steps, and forming a structured methodology.

Therefore, as illustrated in Figure 1, in our **CogMath** framework, we evaluate LLMs' mathematical abilities from the above three stages of problem-solving process. For each stage, we design multiple dimensions to assess LLMs from various perspectives. For instance, to assess a model's *problem comprehension* stage, beyond investigating its accuracy after rephrasing the original problem, we can explore its sensitivity to changes in problem conditions, such as adding irrelevant information or disrupting problem sentences. Overall, for these three stages, we develop a total of nine dimensions that form a cohesive and comprehensive evaluation, with details presented in Table 1.

| Stages | Dimensions | Example of Inquiry $q_i$ | Pass |
|---|---|---|---|
| Problem Comprehension | **Dimension 1:** Sentence Paraphrasing | Jacob had $21. Emily shared half of her $100 with him. How much money does Jacob have now? | Answer Correctly |
| | **Dimension 2:** Sentence Disruption | $21 Ali had half of $100 him Leila her gave now? does Ali much How have | Identify "Unsolvable" |
| | **Dimension 3:** Missing Condition | Ali had some money. Leila gave him half of her money. How much does Ali have now? | Identify "Unsolvable" |
| | **Dimension 4:** Redundant Condition | Ali had $21. Leila gave him half of her $100. Before meeting with Leila, Ali had already counted his money twice to make sure it was correct. How much does Ali have now? | Answer Correctly |
| Problem Solving | **Dimension 5:** Analogical Reasoning | Tom had $21 comic books. Jerry traded him half of his collection of $100 comic books. How many comic books does Tom have now? | Answer Correctly |
| | **Dimension 6:** Numerical Transformation | Ali had $30. Leila gave him half of her $120. How much does Ali have now? | Answer Correctly |
| | **Dimension 7:** Knowledge Redefinition | Assume "half" means one-third of the given amount, solve the following problem: Ali had $21. Leila gave him half of her $100. How much does Ali have now? | Answer Correctly |
| Solution Summarization | **Dimension 8:** Intermediate Step Questioning | Given the mathematical problem: Ali had $21. Leila gave him half of her $100. How much does Ali have now? please answer my following question: Why does Ali now have $71? | Answer Correctly |
| | **Dimension 9:** Backward Reasoning | In the problem, "Ali had $21. Leila gave him half of her $\alpha$, where $\alpha$ is an unknown total amount of money Leila had. How much does Ali have now?", if Ali now has $71, what is the value of $\alpha$? | Answer Correctly |

Table 1: The 3 cognitive stages and 9 dimensions in our CogMath. "Pass" refers to the type of LLM response that is considered to pass the inquiry $q_i$ of the given dimension.

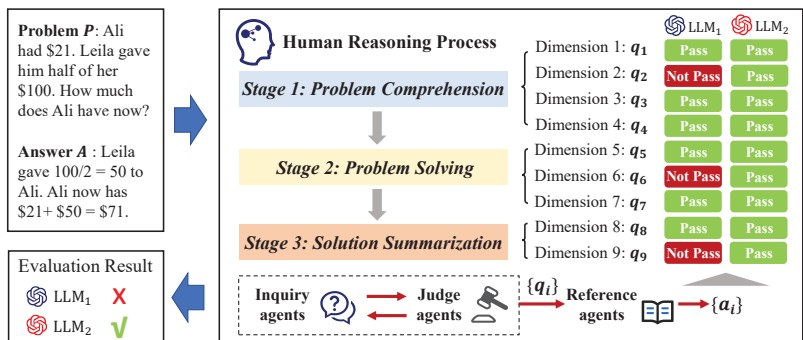

Figure 1: Illustration of our CogMath framework.

In each dimension $i$, to scientifically quantify a LLM's performance, we design an "*Inquiry-Judge-Reference*" multi-agent system. The *Inquiry* agent poses an inquiry $q_i$ related to the original problem $P$ that aligns with the given dimension. The *Judge* agent evaluates the quality of $q_i$ and repeatedly invokes the *Inquiry* agent until a reasonable inquiry is obtained or the maximum number of iterations $\delta$ is reached. The *Reference* agent generates an answer $a_i$ to $q_i$, which is used to assess whether a real LLM's response to $q_i$ is correct. Depending on the dimension, the inquiry $q_i$ could be a question about the solution steps of problem $P$, a rewording of problem $P$ that does not affect its difficulty or required knowledge, or a counterfactual question (e.g., removing a necessary condition) aimed at testing the robustness. Prompts for all agents are presented in Appendix A. For humans, truly mastering a mathematical problem requires a solid performance at each dimension. Hence, in CogMath, only when a LLM passes all dimensions can we conclude that it has genuinely mastered the problem $P$. Notably, these evaluation results also serve as a multifaceted analysis of the model, revealing gaps between its performance in each dimension and human cognition.

### 3.1 STAGE 1: *Problem Comprehension*

The *problem comprehension* stage serves as the foundation for the entire problem-solving process. From fine-grained to coarse-grained understanding, it involves capturing the details of words, phrases, and sentences in the problem, as well as translating the mathematical concepts, conditions, and definitions on a broader scale. To evaluate how well a LLM performs in this stage, we focus on

how it grasps the underlying implications of various perturbations made to the original problem and responds appropriately. For different granularities, we design four dimensions:

• **Dimension 1: Sentence Paraphrasing.** If a human truly understands a mathematical problem, she will demonstrate a robust understanding of the problem's meaning despite changes in wording or sentence structure. Inspired by this, this dimension evaluates the LLM's ability to understand a problem that has been rephrased using different but synonymous expressions. Successful handling of this dimension indicates the model's proficiency in grasping the core concepts and recognizing that, despite linguistic variations, the underlying problem remains unchanged.

To achieve this, we ask the *Inquiry* agent to pose a paraphrased version of the original problem $P$ as $q_1$, while preserving the mathematical essence (e.g., "Jacob had \$21 ..." in Table 1). Since the answer to the rephrased problem remains the same as the original, the *Reference* agent can directly use the original answer as the reference $a_1$ for evaluation. This ensures that the inquiry focuses on how well the model can interpret the reworded problem without needing to solve it differently.

• **Dimension 2: Sentence Disruption.** To prevent a LLM from simply memorizing the solution based on the semantics of the original problem, we propose this dimension from a counterfactual perspective. To disentangle the impact of semantics on reasoning, the *Inquiry* agent randomly disrupts the word order within each clause of the original problem, creating a "pseudo problem" $q_2$, where the words remain the same as in $P$, but from a human perspective, the entire problem is unreadable and unsolvable. In this case, the *Reference* agent does not need to generate an answer, as the expected response is simply "unsolvable", and the *Judge* agent is also no longer required to make any judgments. If the LLM's response to $q_2$ is the same with the original answer, it indicates that this model is likely recalling an answer based on certain keywords or patterns rather than truly understanding the problem. Therefore, this dimension helps us assess whether the LLM is genuinely solving the problem or relying on superficial clues (Sun et al., 2023).

• **Dimension 3: Missing Condition.** For humans, understanding what the given conditions are in a math problem is a critical step in the comprehension process. If essential conditions are missing, we can recognize that the problem becomes unsolvable. Therefore, in this dimension, we still adopt a counterfactual approach: if, after the removal of a necessary condition, the LLM is still able to produce the original answer, it suggests that the model is relying on the semantic similarity to the memorized problem to map out the solution, rather than genuinely solving it. As illustrated in Appendix A.2.1, we ask the *Inquiry* agent to omit one key condition from the original problem, presenting an underspecified version of $P$ as inquiry $q_3$. The *Judge* agent needs to carefully assess whether only one condition has been removed and whether the inquiry $q_3$ does not alter any other parts of the original problem's formulation. The *Reference* agent does not generate an answer, as the model should also recognize that $q_3$ is unsolvable without the missing information.

• **Dimension 4: Redundant Condition.** In contrast to Missing Condition, we design this dimension that introduces irrelevant conditions into the problem. For example, an extra condition such as "Before meeting ... make sure it was correct" might be added to the problem shown in Table 1. A LLM that truly masters problem $P$ should distinguish between essential and non-essential information, ensuring that unnecessary data does not interfere with the reasoning process. Therefore, the *Inquiry* agent presents a problem $q_4$ with one redundant condition. The *Judge* agent evaluates whether the extraneous detail does not affect the solution to the original problem, and the *Reference* agent provides the original answer, as the added information should not affect the solution.

## 3.2 STAGE 2: *Problem Solving*

This stage primarily involves three key components: solving strategy, numerical calculation, and mathematical knowledge (Sweller, 1988; Jonassen, 2000). The solving strategy is an organization of logical thinking specific to the problem, numerical calculation refers to arithmetic operations, and mathematical knowledge reflects common principles that apply across problems. These three components are orthogonal to each other and together form the foundation of reasoning. To evaluate whether a LLM genuinely grasps these components, we design the following three dimensions:

• **Dimension 5: Analogical Reasoning.** The solving strategy serves as a commonality across different problems, allowing a human to solve multiple similar problems using the same underlying logic. To this end, in this dimension, the *Inquiry* agent presents a problem that is conceptually consistent

to, but not identical to, the original problem $P$, as $q_5$ (e.g., "Tom had 21 comic books..." in Table 1). This tests the LLM's ability to generalize the solving strategy, demonstrating a deeper understanding of the underlying reasoning thought. To be notice, $q_5$ does not alter the problem-solving process. It retains the same approach, difficulty level, and required knowledge, with the complexity and core principles remain unchanged. Based on this, the *Reference* agent also does not need to generate a completely new answer. Instead, it makes minor adjustments to the original solution, ensuring the accuracy of the new answer $a_5$ (Appendix A.4.3).

• **Dimension 6: Numerical Transformation.** Generally, the solving strategy represents the essential structure of solution, and the final solving step can be seen as plugging the numerical values from the original problem into the strategy. Therefore, if a human has mastered the problem, changing the numerical values will not affect the ability to solve it. Based on this idea, in this dimension, the inquiry $q_6$ is a variant of the original problem $P$ that modifies its numerical values (e.g., replace the numbers "21" and "100" with "30" and "120" in Table 1). Since $q_6$ is a new problem, we instruct the *Reference* agent to refer to the original answer and provide a corresponding new answer.

• **Dimension 7: Knowledge Redefinition.** Knowledge forms the foundation of human cognition, guiding how abstract principles are applied during the solution process (Goldman, 1986; Habermas, 2015). For example, solving the problem in Figure 1 requires commonsense about the concept of "half". This understanding is flexible—if the problem redefines the calculation of "half", a human who truly grasps the concept will adapt her reasoning to fit the new definition. This process implies that an authentic mastery does not simply rely on memorized facts but can adjust the thought process based on the new knowledge. A model that merely relies on pattern recognition or memorization may fail when faced with new definitions, as it lacks the flexible understanding required to adapt.

To assess if a LLM can do this, the *Inquiry* agent adaptively modifies a key mathematical definition within problem $P$ by introducing a statement like "Assume 'half' means one-third of the given amount" in inquiry $q_7$. This redefinition forces the LLM to adapt its solution based on the modified concept. The *Reference* agent then generates a new solution based on the redefined knowledge, and the *Judge* agent assesses whether $q_7$ with the new definition is solvable.

## 3.3 STAGE 3: *Solution Summarization*

After completing a problem-solving stage, humans often reflect on their reasoning process, summarizing the steps they took and the methodology behind their approach (Cottrell, 2023; Dewey, 2022). This summarization helps consolidate the understanding of not just the solution, but also the overall thought process, which can then be applied to similar problems in the future. In this stage, a human that truly masters the problem can accurately recall intermediate reasoning steps and verify the solution by working backward. To mimic these processes, we examine two critical dimensions:

• **Dimension 8: Intermediate Step Questioning.** In human reasoning, breaking down the problem-solving process into smaller, manageable steps is essential for clarity and learning. Beyond evaluating the final answer, assessing whether a LLM has precisely understood the intermediate steps is an indispensable part of determining if it truly grasps a problem. Therefore, in this dimension, the *Inquiry* agent presents an inquiry $q_8$ that asks a LLM to explain one of the key intermediate steps during the problem-solving process (e.g., step 2 in Appendix A.7.1). This ensures that the model is not just arriving at a correct final answer by coincidence or pattern recognition, but is following a clear, logical sequence throughout the entire solution. Then, the *Judge* agent checks whether $q_8$ corresponds to a specific step in the original reasoning process, and the *Reference* agent generates an explanation for this step based on the original solution.

• **Dimension 9: Backward Reasoning.** Inspired by Yu et al. (2024); Weng et al. (2023), backward reasoning is a crucial and challenging mathematical reasoning ability. It refers to inferring missing information by reasoning backward from the solution, mirroring how humans check their thought by retracing their reasoning to ensure there are no mistakes (Rips, 1994). Therefore, it can be used to evaluate whether LLMs maintain consistency and logical coherence from both directions—forward and backward. If a model truly understands the problem-solving process, it should be able to perform this reverse reasoning without contradictions.

For this purpose, our *Inquiry* agent formulates inquiry $q_9$ by masking a key numerical value from the original problem $P$ and requiring the model to infer the missing value based on the original solution.

| | | MATH | | | | | | | | GSM8K | MExam |
|---|---|---|---|---|---|---|---|---|---|---|---|
| | | Avg | Alg | Count | Geo | Itmd | Num | Pre-Alg | Pre-Cal | | |
| GPT-4 | Vanilla | 0.758 | 0.908 | 0.783 | 0.660 | 0.580 | 0.792 | 0.879 | 0.574 | 0.954 | 0.807 |
| | CogMath | 0.393 | 0.532 | 0.395 | 0.276 | 0.197 | 0.337 | 0.587 | 0.266 | 0.671 | 0.364 |
| | Δ | -0.365 | -0.376 | -0.388 | -0.384 | -0.383 | -0.455 | -0.292 | -0.308 | -0.283 | -0.440 |
| GPT-3.5 | Vanilla | 0.482 | 0.672 | 0.426 | 0.390 | 0.276 | 0.415 | 0.693 | 0.273 | 0.838 | 0.531 |
| | CogMath | 0.176 | 0.280 | 0.108 | 0.121 | 0.062 | 0.109 | 0.315 | 0.088 | 0.424 | 0.192 |
| | Δ | -0.306 | -0.392 | -0.318 | -0.269 | -0.214 | -0.306 | -0.378 | -0.185 | -0.414 | -0.339 |
| Gemini-1.5 | Vanilla | 0.615 | 0.812 | 0.535 | 0.489 | 0.423 | 0.555 | 0.781 | 0.479 | 0.922 | 0.739 |
| | CogMath | 0.291 | 0.428 | 0.247 | 0.173 | 0.142 | 0.206 | 0.455 | 0.205 | 0.500 | 0.338 |
| | Δ | -0.325 | -0.385 | -0.288 | -0.316 | -0.281 | -0.349 | -0.326 | -0.274 | -0.422 | -0.401 |
| Llama3-8B | Vanilla | 0.336 | 0.458 | 0.258 | 0.217 | 0.194 | 0.267 | 0.540 | 0.222 | 0.826 | 0.455 |
| | CogMath | 0.056 | 0.081 | 0.044 | 0.025 | 0.016 | 0.024 | 0.123 | 0.020 | 0.342 | 0.096 |
| | Δ | -0.280 | -0.377 | -0.214 | -0.192 | -0.178 | -0.243 | -0.417 | -0.202 | -0.484 | -0.359 |
| Llama2-13B | Vanilla | 0.106 | 0.142 | 0.080 | 0.073 | 0.051 | 0.074 | 0.196 | 0.059 | 0.446 | 0.267 |
| | CogMath | 0.008 | 0.013 | 0.004 | 0.004 | 0.003 | 0.001 | 0.016 | 0.004 | 0.064 | 0.024 |
| | Δ | -0.098 | -0.129 | -0.076 | -0.069 | -0.048 | -0.073 | -0.180 | -0.055 | -0.382 | -0.243 |
| Mixtral-8x7B | Vanilla | 0.374 | 0.495 | 0.306 | 0.278 | 0.238 | 0.265 | 0.529 | 0.339 | 0.575 | 0.506 |
| | CogMath | 0.092 | 0.147 | 0.053 | 0.058 | 0.037 | 0.028 | 0.165 | 0.079 | 0.212 | 0.133 |
| | Δ | -0.282 | -0.348 | -0.253 | -0.220 | -0.201 | -0.237 | -0.364 | -0.260 | -0.363 | -0.373 |
| Deepseek-v2.5 | Vanilla | 0.747 | 0.915 | 0.730 | 0.597 | 0.548 | 0.780 | 0.870 | 0.625 | 0.951 | 0.855 |
| | CogMath | 0.368 | 0.519 | 0.346 | 0.284 | 0.207 | 0.285 | 0.526 | 0.233 | 0.646 | 0.342 |
| | Δ | -0.379 | -0.396 | -0.384 | -0.313 | -0.341 | -0.495 | -0.344 | -0.392 | -0.305 | -0.513 |

Table 2: Performance of different LLMs on vanilla datasets and our CogMath framework.

The *Reference* agent directly takes the masked value as the answer $a_9$, and the *Judge* agent evaluates whether the masked problem, when combined with the original answer, remains solvable.

## 4 EVALUATION

### 4.1 DATA COLLECTION

To achieve comprehensive evaluation, we apply our CogMath to two of the most representative mathematical benchmarks, GSM8K (Cobbe et al., 2021) and MATH (Hendrycks et al., 2021), along with our constructed MExam dataset. GSM8K is an elementary-level math word problem dataset that primarily involves numerical understanding and reasoning skills. MATH is a high school competition-level dataset, consisting of 7 subcategories, such as algebra, geometry, and number theory. MExam is composed of 6,353 questions manually collected from real exams, which covers the full K-12 mathematics curriculum. For GSM8K and MATH, since their training sets may have already been used in the training process of current LLMs, we apply CogMath on their public test sets, which contain 1,319 and 5,000 questions, respectively.

### 4.2 EXPERIMENTAL SETUP

We evaluated seven mainstream LLMs, including four closed-source models: GPT-4 (Achiam et al., 2023), GPT-3.5-Turbo (OpenAI, 2023), Gemini-1.5-Flash (Team et al., 2023), and Deepseek-v2.5 Liu et al. (2024a), as well as three open-source models: Llama3-8B (Meta, 2024), Llama2-13B (Touvron et al., 2023), and Mixtral-8x7B-Instruct (MistralAITeam, 2023). The implementation details of CogMath are presented in Appendix B. We use *Pass Rate* (PR) as our metric. This is because, in CogMath, dimensions 2 and 3 are based on counterfactual settings. Therefore, for inquiries $q_2$ and $q_3$, the expected response is "unsolvable" (Table 1), and when the LLM's response differs from the original answer, we consider it to have passed the corresponding inquiry. For the remaining seven dimensions and the original dataset, the *Pass Rate* is equivalent to *Answer Accuracy*.

### 4.3 MAIN RESULTS

Table 2 presents the original results ("Vanilla") of all LLMs as well as their performance under our CogMath framework. First, as observed, there is a significant decrease of 30-40% in pass rates for all models, indicating that the mathematical abilities they display on public benchmarks may not be as

| | | MATH | | | | | | | | GSM8K | MExam |
|---|---|---|---|---|---|---|---|---|---|---|---|
| | | Avg | Alg | Count | Geo | Itmd | Num | Pre-Alg | Pre-Cal | | |
| GPT-4 | *Stage 1* | 0.630 | 0.813 | 0.671 | 0.459 | 0.401 | 0.635 | 0.798 | 0.452 | 0.851 | 0.690 |
| | *Stage 2* | **0.532** | **0.683** | **0.534** | **0.395** | **0.323** | **0.485** | **0.728** | **0.401** | 0.870 | 0.624 |
| | *Stage 3* | 0.699 | 0.773 | 0.698 | 0.595 | 0.604 | 0.711 | 0.790 | 0.630 | **0.832** | **0.600** |
| GPT-3.5 | *Stage 1* | 0.359 | 0.561 | 0.283 | 0.246 | **0.147** | 0.257 | 0.571 | **0.194** | **0.636** | 0.443 |
| | *Stage 2* | **0.334** | **0.482** | **0.262** | **0.228** | 0.161 | **0.250** | **0.543** | 0.209 | 0.707 | **0.407** |
| | *Stage 3* | 0.486 | 0.574 | 0.460 | 0.397 | 0.396 | 0.465 | 0.563 | 0.443 | 0.662 | 0.474 |
| Gemini-1.5 | *Stage 1* | 0.509 | 0.715 | 0.428 | 0.388 | 0.307 | 0.415 | 0.692 | 0.372 | 0.829 | 0.618 |
| | *Stage 2* | **0.421** | **0.586** | **0.380** | **0.284** | **0.240** | **0.300** | **0.629** | **0.302** | 0.806 | **0.579** |
| | *Stage 3* | 0.659 | 0.741 | 0.660 | 0.534 | 0.571 | 0.678 | 0.718 | 0.623 | **0.748** | 0.653 |
| Llama3-8B | *Stage 1* | 0.168 | 0.256 | 0.133 | 0.094 | **0.059** | **0.094** | 0.318 | **0.090** | 0.607 | 0.301 |
| | *Stage 2* | **0.160** | **0.215** | **0.118** | **0.079** | 0.079 | 0.106 | **0.307** | 0.104 | 0.626 | **0.294** |
| | *Stage 3* | 0.303 | 0.356 | 0.314 | 0.240 | 0.235 | 0.244 | 0.392 | 0.267 | **0.556** | 0.348 |
| Llama2-13B | *Stage 1* | **0.039** | 0.063 | 0.076 | **0.019** | **0.019** | 0.012 | 0.085 | **0.011** | 0.243 | **0.118** |
| | *Stage 2* | 0.047 | **0.062** | **0.027** | 0.029 | 0.037 | 0.024 | **0.080** | 0.037 | 0.253 | 0.133 |
| | *Stage 3* | 0.117 | 0.132 | 0.122 | 0.081 | 0.113 | 0.094 | 0.140 | 0.103 | **0.232** | 0.289 |
| Mixtral-8x7B | *Stage 1* | **0.200** | **0.308** | **0.131** | **0.127** | **0.094** | **0.113** | **0.327** | **0.150** | **0.400** | 0.364 |
| | *Stage 2* | 0.224 | 0.328 | 0.139 | 0.146 | 0.136 | 0.133 | 0.344 | 0.185 | 0.430 | **0.332** |
| | *Stage 3* | 0.398 | 0.434 | 0.376 | 0.372 | 0.341 | 0.337 | 0.490 | 0.374 | 0.569 | 0.432 |
| Deepseek-v2.5 | *Stage 1* | 0.649 | 0.844 | 0.578 | 0.507 | 0.455 | 0.683 | 0.780 | 0.491 | 0.832 | 0.717 |
| | *Stage 2* | **0.526** | **0.695** | **0.496** | **0.411** | **0.328** | **0.463** | **0.723** | **0.357** | 0.850 | 0.672 |
| | *Stage 3* | 0.681 | 0.762 | 0.692 | 0.610 | 0.607 | 0.644 | 0.741 | 0.623 | **0.817** | **0.541** |

Table 3: Performance of different LLMs at each cognitive stage.

genuine and reliable as they appear. Even GPT-4 successfully passes all dimensions of CogMath on only 39.3% and 67.1% of the problems in MATH and GSM8K datasets, respectively. Second, on the more challenging MATH dataset, the most powerful models (i.e., perform best in "Vanilla"), GPT-4 and Deepseek-v2.5, exhibit the largest drops, with $\Delta$ values of 36.5% and 37.9%, respectively. However, on the simpler GSM8K dataset, their declines are the smallest, with $\Delta$ =28.3% and 30.5%, respectively. This suggests that the extent to which the capabilities of LLMs are overestimated does not diminish as the models become stronger, but rather remains a widespread phenomenon unrelated to model size or dataset difficulty. Third, we observe that the issue of overestimated model capability persists on our newly constructed MExam dataset, which has not been used in the training of these LLMs. On one hand, this suggests that the overestimation of mathematical capabilities is not solely due to data contamination. We posit that one key reason for this phenomenon may be that LLMs primarily capture the superficial pattern of reasoning from training. While this simulation has the potential to generalize and generate correct answers for unseen problems, it does not represent true mastery of mathematical principles, which is fragile and lacks robustness. On the other hand, this phenomenon demonstrates that simply introducing more test problems may be insufficient to assess the true mathematical abilities of LLMs. It is crucial to use our CogMath to scrutinize their performance across various cognitive stages and dimensions of reasoning.

## 4.4 ANALYSIS ON THREE COGNITIVE STAGES

To further analyze the extent to which LLMs grasp different cognitive stages, we present the *Pass Rate* at different stages in Table 3, with the stage having the lowest pass rate highlighted in bold. Specifically, we first observe that for weaker LLMs (e.g., Llama2-13B), their pass rates in Stage 1 (i.e., *problem comprehension*) are the lowest, indicating that these models already exhibit deficiencies in fundamental understanding. For more advanced models (e.g., GPT-4, Deepseek-v2.5), their comprehension abilities appear more stable. Even when a subtle condition is removed or added, these models can still recognize it and determine whether the new problem is unsolvable. However, they still struggle significantly with mastering Stage 2 (i.e., *problem solving*). For instance, on the MATH dataset, GPT-4 and Deepseek exhibit pass rates of only 53.2% and 52.6%, respectively. This further confirms that large models have not yet genuinely mastered the problem-solving process in mathematics, and we find that the main reason is that their grasp of knowledge is still unstable (described in Section 4.5). Finally, the pass rate in Stage 3 (i.e., *Solution Summarization*) remains below 0.85. This suggests that current LLMs are more suited for forward reasoning, i.e., generating answers based on the problems, but struggle to assess whether the solution aligns with the original

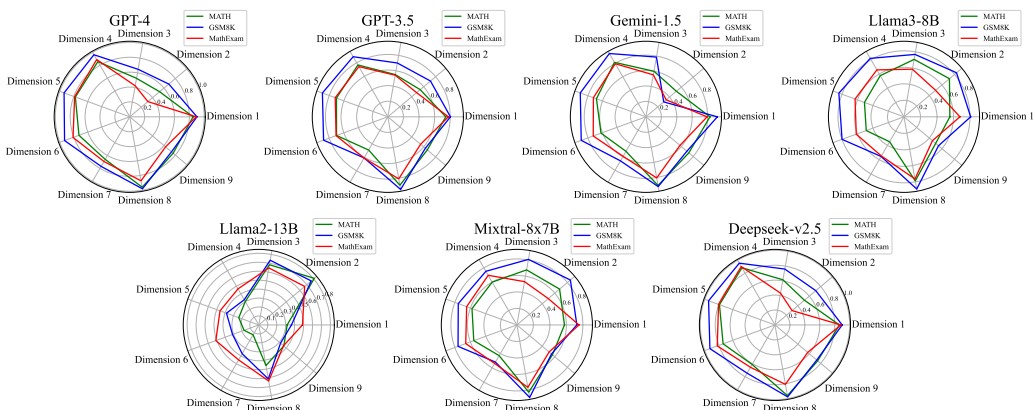

Figure 2: *Relative Pass Rate* (RPR) of different LLMs in each dimension.

problem from a backward perspective. This finding is consistent with existing research that shows LLMs may find it challenging to verify the correctness of their own answers (Huang et al., 2024).

## 4.5 ANALYSIS ON NINE COGNITIVE DIMENSIONS

Furthermore, we analyze the performance of LLMs in each dimension, which allows us to explore in greater detail the model's robustness and sensitivity. Specifically, for each dimension $i$, we calculate a *Relative Pass Rate* (RPR) defined as: RPR = $\frac{|Pass_i \cap Pass|}{|Pass|}$. Here, $Pass_i$ denotes the problems where the LLM successfully passes their corresponding inquiry $q_i$, and $Pass$ refers to the problems correctly answered in the original dataset. It is important to note that a higher RPR indicates better robustness and stability of the LLM's capabilities in that dimension. This is because the model's performance on corresponding inquiries is highly consistent with its performance on the original problems, making it less likely to exhibit defects when it answers the original problem correctly. Conversely, a lower RPR signifies a more detrimental impact on LLM performance, suggesting that the model exhibits lower adaptability to that type of inquiry.

The results are presented in Figure 2. Overall, Deepseek-v2.5 and GPT-4 exhibit the most balanced performance across multiple dimensions, followed by GPT-3.5, Mixtral-8x7B, Gemini-1.5, and Llama3-8B, with Llama2-13B performing the worst. Secondly, regarding the four dimensions in *problem comprehension* stage, an important observation is that GPT-4, GPT-3.5, Gemini-1.5, and Deepseek-v2.5 underperform in Dimensions 2 and 3, even lagging behind Llama2-13B and Llama3-8B. We speculate that this is because most training data for current LLMs is composed of solvable math problems. After being trained on such data, when facing an unsolvable problem, current LLMs may inherently "over-correct" the problem into a solvable one by rephrasing, adding conditions, or reorganizing, aligning it more closely with their training data. This insight suggests that in order to equip LLMs with more human-like cognitive capabilities, it is necessary to cultivate critical thinking skills rather than mere imitation of training data. Thirdly, for the three dimensions associated with *problem solving* stage, Dimension 7 accounts for the low pass rate discussed in Section 4.4. This indicates that current LLMs treat knowledge more as rigid memorization and application, rather than integrating it organically and flexibly into the reasoning process. Lastly, in *solution summarization* stage, nearly all LLMs demonstrate higher RPR values in Dimension 8, suggesting that they are quite adept at explaining reasoning steps. However, the performance in Dimension 9 indicates that these models struggle to use conclusions to reversely derive conditions, which explain why they are difficult to self-verify the correctness of their own answers.

## 4.6 EFFECT OF REASONING ENHANCEMENT METHODS

To analyze the impact of different reasoning enhancement methods on LLMs' mathematical abilities, we explore two commonly used prompting techniques in this section: Chain-of-Thought (CoT) (Wei et al., 2022) and In-Context Learning (ICL) (Dong et al., 2022). For CoT, we prompt the LLM to answer each inquiry in CogMath "step by step". For ICL, we adopt a one-shot setting where, for each dimension $i$, we randomly sample a problem $P_i$ from the training set and use CogMath to construct an (inquiry $q_P^i$, answer $a_P^i$) pair as the demonstration.

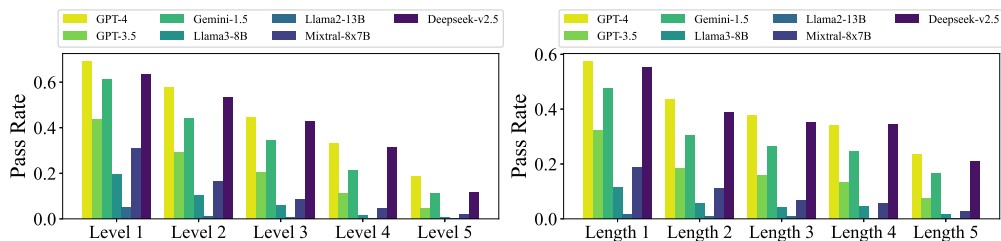

Figure 3: Relationship between LLM performance with problem characteristics.

|  |  | Vanilla | CogMath | CogMath(CoT) | CogMath(ICL) |
|---|---|---|---|---|---|
| MATH | GPT-4 | 0.758 | 0.393 | 0.380 | 0.368 |
|  | GPT-3.5 | 0.482 | 0.176 | 0.169 | 0.167 |
|  | Gemini-1.5 | 0.615 | 0.291 | 0.242 | 0.250 |
| GSM8K | GPT-4 | 0.964 | 0.671 | 0.680 | 0.676 |
|  | GPT-3.5 | 0.531 | 0.424 | 0.442 | 0.466 |
|  | Gemini-1.5 | 0.739 | 0.500 | 0.585 | 0.518 |

Table 4: Performance of different reasoning enhancement methods.

As shown in Table 4, these techniques led to a performance decrease of 0.7% (0.176 → 0.169) to 4.9% (0.291 → 0.242) on MATH dataset but an increase of 0.5% (0.671 → 0.676) to 8.5% (0.500 → 0.585) on GSM8K. These results suggest that prompting techniques may not fundamentally enhance the mathematical reasoning abilities of large models. Instead, they serve more as an auxiliary tool. For instance, CoT encourages more detailed stepwise reasoning, while ICL focuses on learning and imitating the demonstration. This auxiliary effect may be more effective for simpler datasets like GSM8K, but for more complex problems like those in MATH, since these techniques do not essentially improve the model's capabilities, it is difficult for them to have a positive effect. In some cases, the imitation required by ICL might even limit the model's problem-solving flexibility.

### 4.7 ERROR ANALYSIS

From Sections 4.3 to 4.5, we verify that the primary reason for LLMs making errors in our CogMath is due to their deficiencies in abilities corresponding to Dimensions 2, 3, 7, and 9. In this section, we further investigate how the characteristics of the problems influence LLMs' errors. Specifically, we take the MATH dataset as an example and explore the relationship between problem difficulty and the pass rate, as well as between problem length and the pass rate. Problem difficulty is measured by the dataset's inherent "level" labels, which include five tiers. For problem length, we divide all problems into five levels using an equal-frequency binning approach.

From Figure 3, we can first observe that as problem difficulty increases, the performance of all LLMs declines significantly. More specifically, most models only perform well on level 1 problems, while only GPT-4 and Deepseek demonstrate proficiency on more than half of the problems at both levels 1 and 2. Secondly, as problem length increases, the LLM performance also shows some decline, though it is less significant compared to the impact of problem difficulty. This suggests that problem length has a relatively lower correlation with model performance. Based on these observations, we think future improvements in LLMs' mathematical abilities could focus on enhancing their capacity to handle more complex problems, particularly those in higher difficulty levels.

### 5 CONCLUSION AND DISCUSSION

In this paper, we introduced CogMath, a comprehensive and scientific evaluation framework that assesses the mathematical abilities of large language models across three cognitive stages and nine dimensions of humans. The findings indicated that the mathematical capabilities of current mainstream LLMs are overestimated by approximately 30-40%. Specifically, weaker LLMs like Llama2-13B struggled with problem comprehension, while more advanced LLMs like GPT-4 demonstrated an insufficient grasp of knowledge during problem-solving. Moreover, we verified that prompting techniques such as CoT and ICL do not genuinely enhance the mathematical proficiency of these models. For future work, we discuss some valuable directions in Appendix C.

**Reproducibility Statement.** Our code and data is available at `https://anonymous.4open.science/r/CogMath-2743`, where we provide 100 sample problems along with their corresponding inquiries $q_i$ (and answers $a_i$) from the 9 dimensions in CogMath. Besides, we publish all problems in MExam dataset. All code and data will be publicly available after the paper is accepted.

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

## A  PROMPTS IN COGMATH FRAMEWORK

The prompts for all agents across the 9 dimensions are presented in Figures A.1.1 to A.8.2. Notably, in CogMath, the expected answers for Dimensions 1 to 4 are the original answers $A$ of problem $P$, so we omit the corresponding *Reference* agents for these dimensions. For Dimension 2, the *Inquiry* agent automatically disrupts the word order in each clause according to rules, and this process does not require a special prompt or a *Judge* agent for evaluation. Hence, all agents for Dimension 2 are omitted here. As for Dimension 9, as shown in Figure A.8.1, its *Inquiry* agent also automatically determines the answer for inquiry $q_9$ (marked with "[]"), so there is no need to design an additional *Reference* agent prompt, which is therefore omitted.

## B  IMPLEMENTATION DETAILS.

All the *Inquiry* agents, *Reference* agents, and *Judge* agents are implemented with GPT-4. Besides, the maximum number of iterations for *Inquiry* agent is set to $\delta = 10$. If after 10 iterations, we still fail to obtain a satisfactory inquiry, we consider the problem to be unsuitable to be evaluated from that dimension. For such problems, we omit consideration of that dimension during the evaluation.

## C  DISCUSSION

First, our CogMath framework is highly generalizable, as it does not rely on specific problem types or formats, making it applicable to testing LLMs' cognitive abilities in other mathematical tasks, such as theorem proving. Second, our framework can be easily extended to tasks in other domains. For instance, in visual reasoning tasks, a *visual comprehension* stage could be added into our framework, along with dimensions like image perturbation to evaluate the capabilities and robustness of visual LLMs like GPT-4v. Third, through experiments in Sections 4.3 to 4.7, we have conducted a detailed examination of LLMs' mastery across different dimensions, providing valuable insights for future model improvements. For example, as observed in Section 4.5, existing LLMs may exhibit an "over-correction" behavior when faced with unsolvable problems. To address this, we need to introduce critical thinking mechanisms that enable them to reconsider the fundamental nature of each problem, rather than merely imitating patterns from training data. Lastly, from the results of Section 4.6, we found that CoT and ICL may not fundamentally improve the mathematical capabilities of LLMs. However, these techniques have been shown to enhance performance in many NLP tasks. Therefore, we believe that understanding the underlying mechanisms of these methods from a theoretical perspective remains a critical research question.

702
703
704
705
706
707
708
709
710
711
712
713
714
715
716
717
718
719
720
721
722
723
724

### A.1.1: Dimension 1 (*Inquiry* agent) prompt

Now you are a question rewriting agent with interleaving Thought, Action. Thought can be the process of rewriting. Action MUST BE THE REWRITE QUESTION WHICH REPHRASES THE QUESTION. You will be provided with a math problem. Please rephrase the question in a different way.
Here are an example:
Question: Gloria is shoe shopping when she comes across a pair of boots that fit her shoe budget. However, she has to choose between the boots and two pairs of high heels that together cost five dollars less than the boots. If one pair of heels costs $33 and the other costs twice as much, how many dollars are the boots.
Thought: Identify the main elements (character Gloria, items boots and high heels, price relationship), analyze the price relationship (high heels cost $33 and $66, total $99 is $5 less than boots, making boots $104), find the logical relationship (choice, price relationship, and calculation), and change the character (to Alice) and item names (to sneakers and sandals) to rewrite the problem while keeping the prices and relationships the same.
Action: Alice is shopping for footwear when she finds a pair of sneakers that fit her budget. However, she has to choose between the sneakers and two pairs of sandals that together cost five dollars less than the sneakers. If one pair of sandals costs $33 and the other costs twice as much, how many dollars are the sneakers?

Now, here is your question:
Question: {Here is the original problem P}

725
726
727
728
729
730
731
732
733
734
735
736
737
738
739
740
741
742
743
744
745
746
747
748
749
750
751
752
753
754
755

### A.1.2: Dimension 1 (*Judge* agent) prompt

Now you are a judge agent with interleaving Thought, Action. Your task is to determine if the rewritten question is a rephrased version of the original question. Thought can be articulating the logical relationship between the original question and the rewritten question, and analyze whether the logical relationship between the two is consistent. Action must be Yes or No.
Here are an example:
Original question: Janet's ducks lay 16 eggs per day. She eats three for breakfast every morning and bakes muffins for her friends every day with four. She sells the remainder at the farmers' market daily for $2 per fresh duck egg. How much in dollars does she make every day at the farmers' market?
Rewritten question: Kevin's chickens lay 16 eggs each day. He consumes three for his morning meal and uses four to make pastries each day for his neighbors. The remaining eggs are sold at the local farmers' market daily for $2 per egg. How much money does Kevin earn from egg sales each day at the market?
Thought:
Original Question Key Elements:
Subject: Janet's ducks
Daily egg production: 16 eggs
Daily consumption: 3 eggs for breakfast, 4 eggs for muffins
Selling price: $2 per egg
Question focus: Daily earnings from selling eggs at the farmers' market

Rewritten Question Key Elements:
Subject: Kevin's chickens
Daily egg production: 16 eggs
Daily consumption: 3 eggs for morning meal, 4 eggs for pastries
Selling price: $2 per egg
Question focus: Daily earnings from selling eggs at the farmers' market

the rewritten question is a rephrased version of the original question. Both questions convey the same information and ask the same type of question, with only the subject (Janet's ducks vs. Kevin's chickens) and the specific uses of the eggs (breakfast vs. morning meal, muffins vs. pastries) being slightly different. The logical relationship between the two questions is consistent.
Action:Yes

Now, here are your raw question and rewritten question:
Original question:{Here is the original problem P}
Rewritten question:{Here is the inquiry $q_1$}

### A.2.1: Dimension 3 (*Inquiry* agent) prompt

Now you are a question rewriting agent with interleaving Thought, Action. Thought can reason about the necessary conditions for the question. Action MUST BE THE REWRITE QUESTION WHICH REMOVE THE NECESSARY CONDITION.

You will be provided with a math problem. Please analyze the necessary conditions, remove one necessary condition, and make the problem unsolvable.

Here are an example:

Question: Gloria is shoe shopping when she comes across a pair of boots that fit her shoe budget. However, she has to choose between the boots and two pairs of high heels that together cost five dollars less than the boots. If one pair of heels costs $33 and the other costs twice as much, how many dollars are the boots?

Thought: Necessary conditions are

1.Gloria has to choose between purchasing a pair of boots or two pairs of high heels.

2.The price of one pair of high heels is $33.

3.The price of the other pair of high heels is twice the price of the first pair, which is $66.

Action: Gloria is shoe shopping when she comes across a pair of boots that fit her shoe budget. However, she has to choose between the boots and two pairs of high heels. If one pair of heels costs $33 and the other costs twice as much, how many dollars are the boots?

Now, here is your question:

Question:{Here is the original problem P}

### A.2.2: Dimension 3 (*Judge* agent) prompt

Now you are a judge agent with interleaving Thought, Action. Your task is to determine if the rewritten question lacks a crucial condition compared to the original question. Thought can be the comparison of the key conditions of two questions. Action must be Yes or No.

Here are an example:

Original question: Janet's ducks lay 16 eggs per day. She eats three for breakfast every morning and bakes muffins for her friends every day with four. She sells the remainder at the farmers' market daily for $2 per fresh duck egg. How much in dollars does she make every day at the farmers' market?

Rewritten question: Janet's ducks lay 16 eggs per day. She eats three for breakfast every morning and bakes muffins for her friends every day with four. She sells the remainder at the farmers' market daily. How much in dollars does she make every day at the farmers' market?

Thought: Original Question Key Conditions:

Janet's ducks lay 16 eggs per day.

She eats three eggs for breakfast every morning.

She bakes muffins with four eggs every day.

She sells the remainder at the farmers' market daily for $2 per fresh duck egg.

Rewritten Question Key Conditions:

Janet's ducks lay 16 eggs per day.

She eats three eggs for breakfast every morning.

She bakes muffins with four eggs every day.

She sells the remainder at the farmers' market daily.

The rewritten question lacks the crucial condition specifying the price per egg sold at the farmers' market ($2 per fresh duck egg), which is essential for calculating the daily earnings.

Action: Yes

Now, here are your raw question and rewritten question:

Original question:{Here is the original problem P}

Rewritten question:{Here is the inquiry $q_3$}

### A.3.1: Dimension 4 (*Inquiry* agent) prompt

Now you are a question rewriting agent with interleaving Thought, Action. Thought can reason about the unnecessary condition for the question. Action MUST BE THE REWRITE QUESTION WHICH ADDS A UNNECESSARY CONDITION.

You will be provided with a math problem. Please analyze the necessary conditions, add one unnecessary condition, and make the problem solvable.

Here are an example:

Question: Gloria is shoe shopping when she comes across a pair of boots that fit her shoe budget. However, she has to choose between the boots and two pairs of high heels that together cost five dollars less than the boots. If one pair of heels costs $33 and the other costs twice as much, how many dollars are the boots.
Thought: The added irrelevant condition: Gloria's friend informed her that the boots are on sale for a 10% discount, but this information is incorrect and does not affect the actual price of the boots.
Action: Gloria is shoe shopping when she comes across a pair of boots that fit her shoe budget. However, she has to choose between the boots and two pairs of high heels that together cost five dollars less than the boots. If one pair of heels costs $33 and the other costs twice as much, how many dollars are the boots. Additionally, Gloria's friend told her that the boots are on sale for 10% off, but this information is incorrect and does not affect the actual price of the boots.

Now, here is your question:
Question:{Here is the original problem P}

## A.3.2: Dimension 4 (*Judge* agent) prompt

Now you are a judge agent with interleaving Thought, Action. Your task is to determine if the rewritten question has an additional condition compared to the original question. Thought can be the comparison of the key conditions of two questions. Action must be Yes or No.
Here are an example:
Original question: Janet's ducks lay 16 eggs per day. She eats three for breakfast every morning and bakes muffins for her friends every day with four. She sells the remainder at the farmers' market daily for $2 per fresh duck egg. How much in dollars does she make every day at the farmers' market?
Rewritten question: Janet's ducks lay 16 eggs per day. She eats three for breakfast every morning and bakes muffins for her friends every day with four. She sells the remainder at the farmers' market daily for $2 per fresh duck egg. Additionally, she uses exactly two eggs every Sunday to make a special omelette for her family, but this does not affect her daily revenue. How much in dollars does she make every day at the farmers' market?"
Thought:
Original Question Key Conditions:
Janet's ducks lay 16 eggs per day.
She eats three eggs for breakfast every morning.
She bakes muffins with four eggs every day.
She sells the remainder at the farmers' market daily for $2 per fresh duck egg.

Rewritten Question Key Conditions:
Janet's ducks lay 16 eggs per day.
She eats three eggs for breakfast every morning.
She bakes muffins with four eggs every day.
She sells the remainder at the farmers' market daily for $2 per fresh duck egg.
Additionally, she uses exactly two eggs every Sunday to make a special omelette for her family, but this does not affect her daily revenue.

the rewritten question has an additional condition regarding the use of two eggs every Sunday for a special omelette, which is not present in the original question.
Action: Yes

Now, here are your raw question and rewritten question:
Original question:{Here is the original problem P}
Rewritten question:{Here is the inquiry $q_4$}

## A.4.1: Dimension 5 (*Inquiry* agent) prompt

Now you are a question rewriting agent. Please modify the context of the question to test a student's ability to apply their knowledge in different scenarios. While modifying the context, you must not change the solution approach or the specific numerical values in the problem.

Here are an example:
Question: Janet's ducks lay 16 eggs per day. She eats three for breakfast every morning and bakes muffins for her friends every day with four. She sells the remainder at the farmers' market daily for $2 per fresh duck egg. How much in dollars does she make every day at the farmers' market?
Rewritten Question: David's apple trees produce 16 apples per day. He eats three for lunch every afternoon and uses four to make apple pies for his neighbors each day. He sells the remainder at the local grocery store daily for $2 per fresh apple. How much in dollars does he make every day at the grocery store?

Now, here is your question:
Question:{Here is the original problem P}

### A.4.2: Dimension 5 (*Judge* agent) prompt

Now you are a judge agent with interleaving Thought, Action. Your task is to determine if the rewritten question uses the same knowledge points of the original question and only changes the application scene. Thought can be articulating the logical relationship between the original question and the rewritten question, and analyze their knowledge points and application scene. Action must be Yes or No.
Here are an example:
Original question: Janet's ducks lay 16 eggs per day. She eats three for breakfast every morning and bakes muffins for her friends every day with four. She sells the remainder at the farmers' market daily for $2 per fresh duck egg. How much in dollars does she make every day at the farmers' market?
Rewritten question: David's apple trees produce 16 apples per day. He eats three for lunch every afternoon and uses four to make apple pies for his neighbors each day. He sells the remainder at the local grocery store daily for $2 per fresh apple. How much in dollars does he make every day at the grocery store?
Thought:
The solutions of the two question are similar, and both of them only use the basic knowledge of addition and subtraction and income formula. The original question is the application scene where Janet sells duck eggs. The rewritten question is the application scene where David sells apples, so their scenes are different.
Action: Yes

Now, here are your raw question and rewritten question:
Original question:{Here is the original problem P}
Rewritten question:{Here is the inquiry $q_5$}

### A.4.3: Dimension 5 (*Reference* agent) prompt

Now you are a solver agent with interleaving Thought, Action. Your task is to generate the New Answer for the New Question based on the Original Answer of the Original Question. Thought can be to refer to each step of the Original Answer.
Here are an example:
Original Question: Janet's ducks lay 16 eggs per day. She eats three for breakfast every morning and bakes muffins for her friends every day with four. She sells the remainder at the farmers' market daily for $2 per fresh duck egg. How much in dollars does she make every day at the farmers' market?
Original Answer: Janet sells 16 - 3 - 4 = <<16-3-4=9>>9 duck eggs a day.\nShe makes 9 * 2 = $<<9*2=18>>18 every day at the farmer\u2019s market.\n#### 18
New Question: Tom's lemon trees yield 16 lemons per day. He drinks juice made from three for breakfast every morning and uses four to prepare lemonade for his co-workers each day. He sells the remainder at the local outdoor market daily for $2 per fresh lemon. How much in dollars does he make every day at the market?
Thought:
The context changing from eggs to lemons and Janet to Tom.
Action: Tom lefts 16 - 3 - 4 = <<16-3-4=9>>9 lemons a day.\nHe makes 9 * 2 = $<<9*2=18>>18 every day at the local outdoor market.\n#### 18

Now, here are your raw question and rewritten question:
Original Question:{Here is the original problem P}
Original Answer:{Here is the original answer A}
New Question:{Here is the inquiry $q_5$}

### A.5.1: Dimension 6 (*Inquiry* agent) prompt

Now you are a question rewriting agent. Please change the numerical values in the problem, but during the modification, you must not alter the solution approach or the specific context of the problem. The modified values should still be consistent with the meaning of the original problem.
Here are an example:
Question: If a snack-size tin of peaches has $40$ calories and is $2\\\%$ of a person\'s daily caloric requirement, how many calories fulfill a person\'s daily caloric requirement?
Rewritten Question: If a snack-size tin of peaches has $60$ calories and is $3\\\%$ of a person\'s daily caloric requirement, how many calories fulfill a person\'s daily caloric requirement?

Now, here is your question:
Question:{Here is the original problem P}

### A.5.2: Dimension 6 (*Judge* agent) prompt

Now you are a judge agent with interleaving Thought, Action. Your task is to determine if the rewritten question only changes the numbers in the original question. Thought can be articulating the logical relationship between the original question and the rewritten question, and analyze whether their difference is only in numbers. Action must be Yes or No.
Here are two examples:
Original question: Three vertices of a cube in space have coordinates $A = (2,3,0),$ $B = (0,5,4),$ and $C = (4,1,8).$ Compute the coordinates of the center of the cube.
Rewritten question: Three vertices of a cube in space have coordinates \\(A = (3, 2, 1),\\ B = (1, 4, 5),\\) and \\C = (5, 0, 9).\\ Compute the coordinates of the center of the cube.
Thought:
The difference between Original Question and Rewritten question is the coordinates of three points A,B and C. Thus, their difference is only in numbers.
Action: Yes

Original question: John adopts a dog. He takes the dog to the groomer, which costs $100. The groomer offers him a 30% discount for being a new customer. How much does the grooming cost?
Rewritten question: John adopts a cat. He takes the cat to the groomer, which costs $120. The groomer offers him a 25% discount for being a new customer. How much does the grooming cost?
Thought:
The Rewritten question not only changes the number $100 and 30%, but also change "dog" to "cat", which change the meaning of the Original question.
Action: No

Now, here are your raw question and rewritten question:
Original question:{Here is the original problem P}
Rewritten question:{Here is the inquiry $q_6$}

### A.5.3: Dimension 6 (*Reference* agent) prompt

You are a math expert. Please refer to the Original Answer of the Original Question to generate the answer of the New Question.
Original Question: {Here is the original problem P}
Original Answer: {Here is the original answer A}
New Question: {Here is the inquiry $q_6$}

### A.6.1: Dimension 7 (*Inquiry* agent) prompt

Now you are a question rewriting agent. Please redefine some mathematical concepts within the problem to test a student's learning outcomes. For a mathematical concept in the problem, you can change its definition. For example, you can redefine the formula for perimeter or area, but during the redefinition, do not change the original values or context of the problem.
Here are an example:
Question: You draw a rectangle that is 7 inches wide. It is 4 times as long as it is wide. What is the area of the rectangle?
Rewritten Question: Assume the area formula of a rectangle is the sum of its length and width, solve the following problem: You draw a rectangle that is 7 inches wide. It is 4 times as long as it is wide. What is the area of the rectangle?

Now, here is your question:
Question:{Here is the original problem P}

972
973
974
975
976
977
978
979
980
981
982
983
984
985
986
987
988
989
990
991
992
993
994
995
996
997
998
999
1000
1001
1002
1003
1004
1005
1006
1007
1008
1009
1010
1011
1012
1013
1014
1015
1016
1017
1018
1019
1020
1021
1022
1023
1024
1025

### A.6.2: Dimension 7 (*Judge* agent) prompt

Now you are a judge agent with interleaving Thought, Action. Your task is to determine if the two math problems use relatively close knowledge points (we allow differences in the definition or formula used in the solution process). Thought can involve articulating the logical relationship between the two math problems and analyzing their knowledge points. Action must be Yes or No.

Here are an example:

Original question: A sphere is inscribed inside a hemisphere of radius 2. What is the volume of this sphere?

Rewritten question: Assuming the volume of a sphere is calculated by twice the cube of the radius, rather than using the factor $\frac{4}{3}\pi$, solve the problem: A sphere is inscribed inside a hemisphere of radius 2. What is the volume of this sphere?

Thought:

Both problems deal with calculating the volume of a sphere, but the rewritten problem uses a modified formula for the volume (twice the cube of the radius instead of the standard $\frac{4}{3}\pi r^3$). While the specific formula is altered, the core knowledge point—understanding the volume of a sphere and its relationship to the radius—is the same.

Action: Yes

Now, here are your raw question and rewritten question:

Original question:{Here is the original problem P}

Rewritten question:{Here is the inquiry $q_7$}

### A.6.3: Dimension 7 (*Reference* agent) prompt

Please solve the following problem based on its assumption step by step: {Here is the inquiry $q_7$}

### A.7.1: Dimension 8 (*Inquiry* agent) prompt

Now you are a questioning agent with interleaving Thought, Action. Thought can choose one of the steps in the problem reasoning process. Action MUST BE A QUESTION ABOUT THE STEP.

You will be provided with a math problem and its reasoning process. Please choose a step, and ask a question about this step.

Here are an example:

Question: Gloria is shoe shopping when she comes across a pair of boots that fit her shoe budget. However, she has to choose between the boots and two pairs of high heels that together cost five dollars less than the boots. If one pair of heels costs \$33 and the other costs twice as much, how many dollars are the boots.

Reasoning Process: The second pair of heels costs 33 * 2 = \$<<33*2=66>>66.\nThe heels together cost 66 + 33 = \$<<66+33=99>>99.\nThe boots cost \$5 more than both pairs of heels together, so the boots cost 99 + 5 = \$104.\n#### 104

Thought: The reasoning process consists of three steps, choose the second step that calculates the cost of heels.

Action: Why do the heels together cost 99.

Now, here is your question:

Question:{Here is the original problem P}

### A.7.2: Dimension 8 (*Judge* agent) prompt

Now you are a judge agent with interleaving Thought, Action. Your task is to determine if the given question is a correct question about the reasoning process of the original problem. Thought can be the comparison between the question and the reasoning process of the problem. Action must be Yes or No.

Here are an example:

Original question: Janet's ducks lay 16 eggs per day. She eats three for breakfast every morning and bakes muffins for her friends every day with four. She sells the remainder at the farmers' market daily for $2 per fresh duck egg. How much in dollars does she make every day at the farmers' market?
Reasoning process: Janet sells 16 - 3 - 4 = <<16-3-4=9>>9 duck eggs a day.\nShe makes 9 * 2 = $<<9*2=18>>18 every day at the farmer\u2019s market.\n#### 18.
Given question: Why does Janet sell exactly 9 duck eggs a day?
Thought:
Steps in Reasoning process:
Janet sells 9 duck eggs a day.
She makes 18 every day.

The given question asks why does Janet sell 9 duck eggs a day, which is coincident with the reasoning process because the first step explains that Janet sells 9 duck eggs a day.
Action: Yes

Now, here are your raw question and rewritten question:
Original question:{Here is the original problem P}
Reasoning process:{Here is the original answer A}
Given question:{Here is the inquiry $q_8$}

### A.7.3: Dimension 8 (*Reference* agent) prompt

You are a math expert. Please answer my question about the mathematical problem based on the solution: {Here is the original problem P}
Solution: {Here is the original answer A}
My question is: {Here is the inquiry $q_8$}

### A.8.1: Dimension 9 (*Inquiry* agent) prompt

Now you are a question generating agent with interleaving Thought, Action. Thought can choose one of the numeric value in the problem and express it explicitly with []. Action MUST BE A QUESTION THAT MASKS THE NUMERIC VALUE AND ASKS TO DERIVE THE NUMERIC VALUE.
You will be provided with a math problem and its reasoning process. Please choose a numeric value from the problem, mask it with an unknown Greek letter, and generate a question that asks to derive the numeric value.
Here are an example:
Question: Gloria is shoe shopping when she comes across a pair of boots that fit her shoe budget. However, she has to choose between the boots and two pairs of high heels that together cost five dollars less than the boots. If one pair of heels costs $33 and the other costs twice as much, how many dollars are the boots.
Reasoning Process: The second pair of heels costs 33 * 2 = $<<33*2=66>>66.\nThe heels together cost 66 + 33 = $<<66+33=99>>99.\nThe boots cost $5 more than both pairs of heels together, so the boots cost 99 + 5 = $104.\n#### 104
Thought: Mask the number \"five\" in the sentence \"cost five dollars less than the boots\". The value of this number is [5].
Action: For problem \"Gloria is shoe shopping when she comes across a pair of boots that fit her shoe budget. However, she has to choose between the boots and two pairs of high heels that together cost x dollars less than the boots, where x is an unknown value. If one pair of heels costs $33 and the other costs twice as much, how many dollars are the boots.\", we know the answer for this problem is 104, find the value of x.

Now, here is your question:{Here is the original problem P}

### A.8.2: Dimension 9 (*Judge* agent) prompt

Now you are a judge agent with interleaving Thought, Action. Your task is to determine 1) whether the given question does not change the structure of the original question except that an unknown variable is introduced, 2) whether the given question is solvable, and 3) whether the answer to the question is new_answer. Thought can be the comparison between the question and the original question. Action must be Yes or No.
Here are two examples:

Original question: Janet's ducks lay 16 eggs per day. She eats three for breakfast every morning and bakes muffins for her friends every day with four. She sells the remainder at the farmers' market daily for $2 per fresh duck egg. How much in dollars does she make every day at the farmers' market?

Given question: Assuming Janet sells each duck egg at $\alpha$ dollars, where $\alpha$ is unknown. Given that she sells 9 eggs daily, and makes a total of 18 dollars from these sales, what is the value of $\alpha$ in dollars per egg?

New answer: 2

Thought:

The given question states that Janet sells 9 eggs daily, which is not mentioned in the original question. Therefore, the given question changes the semantics of the original question.

Action: No

Original question: A robe takes 2 bolts of blue fiber and half that much white fiber. How many bolts in total does it take?

Given question: For the problem \"A robe takes $\alpha$ bolts of blue fiber and half that much white fiber. How many bolts in total does it take?\", where $\alpha$ is an unknown value. If the total amount of bolts needed is 3, find the value of $\alpha$.

New answer: 2

Thought:

The given question has the same structure of the original question, with only replacing 2 with unknown valuable $\alpha$. Besides, the given question is solvable. Substitute $\alpha$ with 2, the total amount of bolts needed is still 3. Therefore, the answer to the given question is new answer.

Action: Yes

Now, here are your raw question and rewritten question:

Original question:{Here is the original problem P}

Given question:{Here is the inquiry $q_9$}

New answer: {Here is the new_answer $a_9$}

