# OpenReview forum: "CogMath: Evaluating LLMs' Authentic Mathematical Ability from a Cognitive Perspective"
_ICLR.cc/2025/Conference — ICLR 2025 Conference Withdrawn Submission_

### Official Review · Reviewer_o3Lm · 2024-11-03

**Soundness:** 3
**Presentation:** 3
**Contribution:** 3
**Rating:** 5
**Confidence:** 4

**Summary:**

This paper proposes a benchmark for comprehensively evaluating the mathematical abilities of LLMs by examining three cognitive reasoning stages: problem comprehension, problem solving, and solution summarization. Experiments indicate that we may overestimate the capabilities of current LLMs, primarily due to their excessive imitation of superficial reasoning patterns.

**Strengths:**

1.	A comprehensive and scientific benchmark that deeply investigates the flexible reasoning of LLMs is essential for the community.
2.	The authors consider nine dimensions across problem comprehension, solving, and solution summarization, which aids in identifying the main challenges faced by current models.

**Weaknesses:**

1.	While this work addresses cognitive mathematical dimensions comprehensively, I have a question regarding the motivation. Why do the authors believe that previous works introducing perturbations into existing benchmarks are task-specific?
2.	More details are needed about the dataset construction procedure, including how the judge agent is used to ensure the quality of $q_i$, how multiple reference agents are negotiated to finalize the answer, and which foundation models are utilized behind these agents.
3.	For Figure 2, it would be clearer to replace the dimension index with the dimension name. Additionally, for Figure 3, it would be more straightforward if each group of bars represents the same model.
4.	In Section 4.6, the current experiment uses a one-shot setting. Have the authors considered a nine-shot setting, where each demonstration represents one dimension?
5.	In Section 4.7, how are the five tiers of difficulty defined?
6. In Table 1: Should $21 be 21?

**Questions:**

Please refer to the weaknesses section

---

> ### Author Response · Authors · 2024-11-19
> **Authors' Rebuttal: [1/2]**
>
> $\bf{Q1}$:While this work addresses cognitive mathematical dimensions comprehensively, I have a question regarding the motivation. Why do the authors believe that previous works introducing perturbations into existing benchmarks are task-specific?
>
> $\bf{A1}$: Thanks for your valuable question. We describe some previous works as task-specific because their perturbations to benchmarks are often designed to investigate LLMs' performance on specific tasks. For example, [1] expanded the content of questions in GSM8K to evaluate LLMs' ability to handle the long-text problem solving task. [2] transformed GSM8K's problems into symbolic templates to assess LLMs’ reasoning on symbolic tasks.
>
> Please note that we are not suggesting all existing works are task-specific. For some other previous work, we hold their primary limitation lies in relying on an overall accuracy metric without delving into LLMs’ performance at specific cognitive stages. In contrast, our work deconstructs the problem-solving process into three distinct cognitive stages based on human cognition, providing evaluation of how well LLMs grasp each stage. This deeper analysis offers insights that could help improve their capabilities in a more targeted manner.
>
> [1] Can llms solve longer math word problems better?
>
> [2] Gsm-symbolic: Understanding the limitations of mathematical reasoning in large language models.
>
> $\bf{Q2}$:More details are needed about the dataset construction procedure, including how the judge agent is used to ensure the quality of $q_i$, how multiple reference agents are negotiated to finalize the answer, and which foundation models are utilized behind these agents.
>
> $\bf{A2}$: Thank you for raising this valuable question, and sorry for causing any confusion.
> 1) As described in Section 3, each *inquiry* agent is paired with a *judge* agent to assess the quality of the generated $q_i$ (except for Dimension 2, because it relies on rule-based sentence disruption). For each *judge* agent, we prompt GPT-4 with an example to evaluate the quality of $q_i$. The full prompts used for the *judge* agents are presented in Appendix A. If the *judge* agent determines that the quality of $q_i$ is unsatisfying, it responds with "Action: No" in its output. In such cases, the *inquiry* agent will regenerate $q_i$ until the *judge* agent returns "Action: Yes".
> To assess the effectiveness of the *judge* agent, we invite 5 well-trained annotators to assess the final inquires approved by the *judge* agent, evaluating the extent to which they align with the intended dimensions. The pass rates for each dimension on 500 randomly selected problems are shown as follows (Since Dimension 2 relies on rule-based sentence disruption, there is no need for a *judge* agent), ensuring the quality of our inquiries:
> Dimension |D1|D3|D4|D5|D6|D7|D8|D9
>  -|-|-|-|-|-|-|-|-
> Pass rate (*judge*)|0.984|0.992|0.964|0.986|0.986|0.952|0.990|0.950
>
> 2) Regarding the answer finalization, for each *inquiry* agent, there is only ONE *reference* agent independently generating the answer $a_i$ to its corresponding $q_i$, without involving negotiation among multiple *reference* agents. Specifically, for Dimensions 1–4, the *reference* agent simply uses the original problem’s answer as the correct answer for $q_i$. For Dimension 9, the *reference* agent automatically extracts the masked value from the *inquiry* agent’s output as the answer (lines 346-347). For Dimensions 5–8, we rely on prompt engineering with GPT-4 to generate the answers, and all prompts are provided in Appendix A. To ensure the accuracy of these generated answers, we invite 5 well-trained annotators to evaluate the results of 500 randomly selected problems. As shown in the table below, the answers generated by the *reference* agent achieve a pass rate of 95% across all dimensions, which ensures the quality of the results.
> Dimension |D5|D6|D7|D8
>  -|-|-|-|-
> Pass rate (*reference*)|0.954|0.968|0.952|0.986
>
> 3) As mentioned in Appendix B, all agents are implemented using GPT-4.
>
> $\bf{Q3}$:For Figure 2, it would be clearer to replace the dimension index with the dimension name. Additionally, for Figure 3, it would be more straightforward if each group of bars represents the same model.
>
> $\bf{A3}$:Thanks for your constructive feedback. We appreciate your suggestions and will optimize our figures accordingly.

---

> > ### Author Response · Authors · 2024-11-19
> > **Authors' Rebuttal: [2/2]**
> >
> > $\bf{Q4}$:In Section 4.6, the current experiment uses a one-shot setting. Have the authors considered a nine-shot setting, where each demonstration represents one dimension?
> >
> > $\bf{A4}$: Thanks for your insightful question. In Section 4.6, our goal is to investigate the performance of in-context learning (ICL) under our evaluation framework CogMath. Through comparing it with the zero-shot setting in Section 4.3, we aim to quantify the extent to which ICL enhances the reasoning capabilities of LLMs.
> >
> > To evaluate this, we follow the same methodology of Section 4.3 and test the ICL approach on each dimension independently, which assumes that a LLM must answer all 9 inquiries of a problem correctly to truly demonstrate its mastery of this problem. Since the targets and inquiry types of different dimensions are distinct, we believe the most appropriate choice for each dimension is to select the most relevant example within that dimension as the demonstration, while introducing examples from all 9 dimensions might inadvertently dilute the LLM focus on the target dimension. This could also lead to ambiguity in understanding the context, thereby affecting the fairness and reliability of the evaluation.
> >
> > $\bf{Q5}$:In Section 4.7, how are the five tiers of difficulty defined?
> >
> > $\bf{A5}$: The five tiers of difficulty in Section 4.7 are not defined by us but are published by the original MATH dataset[3]. The authors of the MATH dataset encode a problem’s difficulty level from ‘1’ to ‘5’, following AoPS (aops.com/community/c3158_usa_contests). For more details on how these difficulty levels were determined, we recommend referring to the original paper.
> >
> > [3] Measuring Mathematical Problem Solving with the MATH Dataset
> >
> > $\bf{Q6}$:In Table 1: Should $21 be 21?
> >
> > $\bf{A6}$: The word "$21" in Table 1 represents 21 dollars. Since it is the example from the original GSM8K dataset, we believe no modification is necessary.
> >
> > We sincerely hope our rebuttal can address your concerns.

---

> > > ### Author Response · Authors · 2024-11-27
> > >
> > > We wish to once again express our great appreciation for the time you have taken to review our paper. We would appreciate your feedback on whether your main concerns have been adequately addressed. We truly value your understanding and support, and will carefully revise the paper according to your suggestions. Thank you very much!

---

> ### Comment · Area_Chair_x4XQ · 2024-11-27
>
> Dear Reviewer o3Lm,
>
> Thank you for your efforts reviewing this paper. Can you please check the authors' response and see if your concerns have been addressed? Thank you!

---

> > ### Comment · Reviewer_o3Lm · 2024-11-28
> >
> > Dear Authors,
> >
> > Thank you for your explanation and clarification. I have a follow-up question regarding your statement, "we follow the same methodology of Section 4.3 and test the ICL approach on each dimension independently." Have you tested which dimensions benefit most from one-shot ICL and which dimensions benefit the least for the two math datasets?

---

> > > ### Author Response · Authors · 2024-12-02
> > >
> > > Thanks for your follow-up question. To address your concern, we calculate the Pass Rates on all 9 dimensions of our CogMath after incorporating one-shot ICL (i.e., CogMath(ICL) in Section 4.6). As defined in Section 4.2, the higher Pass Rate represents better performance in the specific dimension.
> > >
> > > Pass Rate (MATH) |D1|D2|D3|D4|D5|D6|D7|D8|D9
> > >  -|-|-|-|-|-|-|-|-|-
> > > GPT-4 (CogMath)|0.737|0.589|0.596|0.732|0.695|0.632|0.611|0.943|0.727
> > > GPT-4 (CogMath(ICL))|0.722|0.557|0.645|0.714|0.638|0.681|0.625|0.930|0.720
> > > Improvement|-1.5%|-3.2%|-4.9%|-1.8%|-5.7%|+4.9%|+1.4%|-1.3%|-0.7%
> > >  -
> > > GPT-3.5 (CogMath)|0.486|0.666|0.700|0.486|0.509|0.457|0.379|0.823|0.556
> > > GPT-3.5 (CogMath(ICL))|0.504|0.582|0.733|0.498|0.494|0.499|0.434|0.832|0.559
> > > Improvement|+1.8%|-8.4%|+3.3%|+1.2%|-1.5%|+4.2%|+5.5%|+0.9%|+0.3%
> > >  -
> > > Gemini-1.5 (CogMath)|0.602|0.569|0.764|0.591|0.603|0.531|0.467|0.887|0.696
> > > Gemini-1.5 (CogMath(ICL))|0.595|0.662|0.730|0.589|0.543|0.545|0.466|0.870|0.619
> > > Improvement|-0.7%|+9.3%|-3.4%|-0.2%|-6.0%|+1.4%|-0.1%|-1.7%|-7.7%
> > >
> > > Pass Rate (GSM8K) |D1|D2|D3|D4|D5|D6|D7|D8|D9
> > >  -|-|-|-|-|-|-|-|-|-
> > > GPT-4 (CogMath)|0.886|0.692|0.657|0.946|0.930|0.921|0.792|0.976|0.828
> > > GPT-4 (CogMath(ICL))|0.889|0.662|0.754|0.943|0.920|0.928|0.801|0.972|0.853
> > > Improvement|+0.3%|-3.0%|+9.7%|-0.3%|-1.0%|+0.7%|+0.9%|-0.4%|+2.5%
> > >  -
> > > GPT-3.5 (CogMath)|0.730|0.741|0.728|0.816|0.833|0.792|0.589|0.899|0.668
> > > GPT-3.5 (CogMath(ICL))|0.778|0.640|0.773|0.802|0.810|0.826|0.592|0.901|0.704
> > > Improvement|+4.8%|-10.1%|+4.5%|-1.4%|-2.3%|+3.4%|+0.3%|+0.2%|+3.6%
> > >  -
> > > Gemini-1.5 (CogMath)|0.773|0.730|0.985|0.821|0.890|0.873|0.672|0.907|0.786
> > > Gemini-1.5 (CogMath(ICL))|0.807|0.763|0.859|0.895|0.873|0.868|0.697|0.861|0.773
> > > Improvement|+3.4%|+3.3%|-12.6%|+7.4%|-1.7%|-0.5%|+2.5%|-4.6%|-1.3%
> > >
> > > From the results on MATH and GSM8K datasets above, we can observe:
> > >
> > > - Dimensions 6 and 7 show consistently stable improvements with the introduction of ICL for ALL LLMs. This suggests that ICL has a significant positive impact on the reasoning abilities in handling numerical transformations and knowledge redefinition.
> > >
> > > - Dimension 5 benefits the least from ICL, and it consistently experiences negative effects. This may be due to the reasoning process in the demonstration diverging from that of the original problem, which could disrupt the model's performance in analogical reasoning (i.e., reasoning that follows the same process as the original problem).
> > >
> > > - For other dimensions, the effect of ICL varies across different models. For instance, in Dimension 3, GPT-4 and GPT-3.5 show improvements, while Gemini-1.5 shows a decline. This highlights the differing robustness and capabilities of various models, providing insights into potential weaknesses and future development directions of different LLMs.
> > >
> > > Overall, these findings again demonstrate that our framework provides a systematic and detailed analysis of LLMs across reasoning settings and dimensions, which offers valuable insight to further optimize the reasoning capabilities of LLMs. Thanks for your valuable comment, and we will add these experiments and discussions in our revised version.

---

### Official Review · Reviewer_rBtj · 2024-11-04

**Soundness:** 2
**Presentation:** 3
**Contribution:** 3
**Rating:** 5
**Confidence:** 3

**Summary:**

This paper proposes a multi-agent framework, CogMath, to evaluate the mathematical abilities of LLMs from a cognitive perspective. CogMath breaks down mathematical problem-solving into 3 stages: problem comprehension, problem solving, and solution summarization with nine evaluation dimensions. CogMath generates test samples cross multiple cognitive perspectives using a multi-agent system and reveals that current LLMs' math capabilities are overestimated, which demonstrates the strengths and weaknesses of different models at different evaluation dimensions.

**Strengths:**

- Comprehensive evaluation across nine dimensions can enhance the current math benchmarks.
- Extensive experiments with multiple representative LLMs demonstrate the limitations of current LLMs on math reasoning capabilities.

**Weaknesses:**

- CogMath uses LLMs to construct test samples and evaluate the model-generated answers across multiple dimensions. However, the correctness of generated test cases and the evaluation quality can be a major concern. It would be helpful to add human evaluation on the generated test samples and the judging process.
- The performance degradation can be a regular case instead of an interpretation of overestimation, as the CogMath test questions can be harder than the original questions after processing across multiple dimensions.

**Questions:**

How can the inquiry agents ensure that they generate good questions that meet the dimension requirements? Is there any filtering process involved?

---

> ### Author Response · Authors · 2024-11-19
> **Authors' Rebuttal**
>
> $\bf{Q1}$: CogMath uses LLMs to construct test samples and evaluate the model-generated answers across multiple dimensions. However, the correctness of generated test cases and the evaluation quality can be a major concern. It would be helpful to add human evaluation on the generated test samples and the judging process.
>
> $\bf{A1}$: Thanks for your constructive suggestions. To address your concerns, we first invite 5 well-trained annotators to evaluate the generated test. The pass rates for each dimension on 500 randomly selected problems are shown below (Since Dimension 2 relies on rule-based sentence disruption, there is no need for additional judge):
> Dimension |D1|D3|D4|D5|D6|D7|D8|D9
>  -|-|-|-|-|-|-|-|-
> Pass rate|0.984|0.992|0.964|0.986|0.986|0.952|0.990|0.950
>
> Then, we also invite 5 annotators to evaluate the judging process carried out by GPT4 in our paper. We randomly select 500 judgement results from GPT-4 and ask the annotators to assess their accuracy. The final results indicate that GPT-4 achieves an evaluation accuracy of $\bf{0.958}$, demonstrating the reliability of our judging process.
>
> $\bf{Q2}$: The performance degradation can be a regular case instead of an interpretation of overestimation, as the CogMath test questions can be harder than the original questions after processing across multiple dimensions.
>
> $\bf{A2}$: Thanks for your valuable comment. We fully agree that maintaining consistent question difficulty is crucial. Therefore, we have carefully designed the dimensions in CogMath to avoid significant increases in difficulty. Specifically, for Dimension 1, this dimension involves only rephrasing the original question synonymously, without introducing new knowledge or contexts, thus keeping the difficulty unchanged. For Dimensions 2 and 3, they focus on testing the model’s behavior under counterfactual scenarios, aiming to evaluate overfitting on unanswerable questions, which is irrelevant to question difficulty. For Dimension 4, we introduce an irrelevant condition, which is often a simple factual statement (as shown in Table 1). This does not involve additional computation or reasoning steps and therefore does not affect difficulty. For Dimension 5, we generate a new problem similar to the original one, where the solving process and logic remain identical, ensuring no change in difficulty. For Dimensions 6 and 7, they involve substituting a numerical value or a knowledge definition in the original problem, but the solution process remains the same, leaving the difficulty level unaltered. For Dimensions 8 and 9, their inquiries are designed based on an intermediate step or numerical value from the original problem, which similarly does not exceed the original difficulty.
>
> In summary, each dimension in CogMath ensures that the difficulty of the questions remains consistent, enabling a fairer evaluation of LLMs. Based on your comments, we will incorporate these discussions in the revised paper.
>
> $\bf{Q3}$: How can the inquiry agents ensure that they generate good questions that meet the dimension requirements? Is there any filtering process involved?
>
> $\bf{A3}$: To ensure that *inquiry* agents generate good questions, we have designed a *judge* agent for each *inquiry* agent to evaluate the quality of its generated outputs. If the *judge* agent determines that the output does not meet the standards of the dimension, we ask the *inquiry* agent to regenerate the question. The prompts for all *inquiry* agents and *judge* agents are presented in Appendix A. Based on the results of $\bf{A1}$, the interaction with the *inquiry-judge* agent provides a significant guarantee of the quality of the generated questions.
>
> We sincerely hope our rebuttal can address your concerns.

---

> > ### Author Response · Authors · 2024-11-27
> >
> > We wish to once again express our great appreciation for the time you have taken to review our paper. We would appreciate your feedback on whether your main concerns have been adequately addressed. We truly value your understanding and support, and will carefully revise the paper according to your suggestions. Thank you very much!

---

> ### Comment · Area_Chair_x4XQ · 2024-11-27
>
> Dear Reviewer rBtj,
>
> Thank you for your efforts reviewing this paper. Can you please check the authors' response and see if your concerns have been addressed? Thank you!

---

### Official Review · Reviewer_WAEw · 2024-11-04

**Soundness:** 2
**Presentation:** 2
**Contribution:** 1
**Rating:** 3
**Confidence:** 3

**Summary:**

This work introduces a CogMath framework that consists of nine agents to evaluate the mathematical reasoning ability of large language models from the perspective of comprehension, problem solving and solution summarization.

Specifically, in the comprehension stage, the agents attempt to rephrase, disrupt (permute word ordering), remove condition and add condition of the original question. In the problem solving stage, the agents attempt to conduct analogical reasoning, numerical transformation and knowledge refinement (reshape the semantics of 'half') of the original question. In the solution summarization stage, agents attempt to question the information in the intermediate steps of the solution, and conduct backward reasoning of the question. The experimental results demonstrate that the abilities of current strong LLMs on GSM8K and MATH are overestimated by 30-40% by the calibration of those agents. Besides, CogMath may not serve as an effective prompt-based reasoning enhancement and the problem difficulty and lengths in MATH are negatively correlated with the pass rates in CogMath.

**Strengths:**

This work includes a CogMath framework that could be helpful in evaluating the math reasoning abilities of LLMs more robustly, and covers several dimensions that may introduce perturbation to the stability of reasoning.

**Weaknesses:**

It is not easy to imagine how a handful of agents included in CogMath can be generalized to more challenging questions. For example, how does the backward reasoning being feasible in mathematical proofs. The knowledge redefinition only limits its scope to 'half' and the questions contain the word, where works like FRoG (Li et al. 2024) includes richer quantifier-based variants of GSM8K. It is not surprised to see that the current figures of LLMs in reasoning is not a stable display, but attempts like one-time numerical transformation might make it more robust, but only marginally. Besides, I didn't find enough evidence regarding efforts to make sure the agents faithfully finish their jobs.

This work also collects MExam. However, I know nothing about it from the contents.


Reference

[1] FRoG: Evaluating Fuzzy Reasoning of Generalized Quantifiers in Large Language Models

**Questions:**

See above.

---

> ### Author Response · Authors · 2024-11-19
> **Authors' Rebuttal: [1/2]**
>
> $\bf{Q1}$:It is not easy to imagine how a handful of agents included in CogMath can be generalized to more challenging questions. For example, how does the backward reasoning being feasible in mathematical proofs.
>
> $\bf{A1}$: Thank you for raising this concern. In fact, among the 9 dimensions we propose, only the one you mentioned might depend on the problem's specific format, while the other 8 dimensions are independent of the type of problems and can easily generalize. For instance, in the four dimensions related to *Problem Comprehension* stage, they focus on the rephrasing of word order, expressions, and conditions. As illustrated in Table 1, these can be directly applied to other types of tasks. For the three dimensions in *Problem Solving* phase, Dimension 5 involves generating similar problems, Dimension 6 involves changing numerical values, and Dimension 7 involves modifying knowledge definitions within the problem. These do not depend on the specific problem type too. For *Solution Summarization* phase, Dimension 8 is suitable for problems where their solutions contain multiple steps, which is broadly applicable. For Dimension 9, yes, it may not apply to theorem proofs, but it is adaptable to any QA-format dataset. With this careful consideration, we believe that our agents can be easily generalized to more complex questions without needing additional prompt modifications.
>
> Additionally, compared with expecting an evaluation framework to cover all possible real-world problems, we think what it is more important is the perspective provided by the evaluation, how it reveals the limitations of LLMs, and the insights it offers for further LLM development. For example, in the work by Zhu[1], the authors designed 5 principles based on psychology to rephrase existing benchmarks. Their experiments showed that LLMs are highly sensitive to problem paraphrasing. However, technically, only 2 out of the 5 principles proposed are indeed suitable for problems without multiple-choice options (e.g., GSM8K). Similarly, the FRoG benchmark you mentioned[2] aims to assess LLMs' reasoning with Generalized Quantifiers, but its construction is more suitable for GSM8K-like datasets (which we also used) and is also not applicable to theorem-proofing task. In our work, we emphasize breaking down the problem-solving process into three stages based on human cognitive experiences. By evaluating LLMs’ performance at each of these stages, we can identify limitations in their capabilities at different cognitive levels. To ensure stable and reliable testing, we propose multiple dimensions for assessment within each stage.
>
> In summary, first, the majority of our agents (8 out of 9 dimensions) are indeed applicable to almost any task, and we do not perceive any specific challenges in their generalization. Second, for both prior work and ours, we think it is unnecessary for an evaluation approach to be universally adaptable to all real-world tasks.
>
> [1] Dynamic Evaluation of Large Language Models by Meta Probing Agents
>
> [2] FROG: Evaluating Fuzzy Reasoning of Generalized Quantifiers in Large Language Models
>
> $\bf{Q2}$:The knowledge redefinition only limits its scope to 'half' and the questions contain the word, where works like FRoG (Li et al. 2024) includes richer quantifier-based variants of GSM8K.
>
> $\bf{A2}$: Sorry for causing this misunderstanding. In our paper, the scope of Knowledge Redefinition is not limited to just the redefinition of the word "half." In Table 7, we use this as a simple example to explain the dimension. As shown in Appendix A6.2, this dimension also involves redefining specialized mathematical knowledge, such as the formula of "volume of a sphere." In our anonymous link https://anonymous.4open.science/r/CogMath-2743, we provide 100 problems, each paired with a corresponding knowledge redefinition result. We can find that they cover a variety of topics ranging from "income formula" to "bolt measurement of fiber," alongside other common-sense and mathematical knowledge. Therefore, the range of variants we consider is indeed more diverse.

---

> ### Author Response · Authors · 2024-11-19
> **Authors' Rebuttal: [2/2]**
>
> $\bf{Q3}$:It is not surprised to see that the current figures of LLMs in reasoning is not a stable display, but attempts like one-time numerical transformation might make it more robust, but only marginally.
>
> $\bf{A3}$:Thanks for your valuable feedback. We agree that the instability of reasoning in LLMs is not surprising in many studies. However, as we highlight in $\bf{A1}$, the key is to understand the underlying reasons for this instability, which is exactly what our paper aims to explore. To this end, we decompose the reasoning process into three cognitive stages and design multiple dimensions to evaluate the performance of LLMs at each stage. From our experiments in Section 4.3-4.5, we find that models like LLaMA2 primarily have not truly mastered the fundamental understanding stage, whereas for models like GPT-4, the main issue lies in the stability of knowledge application during the problem-solving stage. These findings suggest distinct directions for future improvements for different models.
>
> $\bf{Q4}$:Besides, I didn't find enough evidence regarding efforts to make sure the agents faithfully finish their jobs.
>
> $\bf{A4}$:Thanks for your question. Regarding each *inquiry* agent, we have designed a *judge* agent to evaluate the quality of its generated outputs. If the *judge* agent determines that the output does not meet the required quality standards, we ask the *inquiry* agent to regenerate the response. To assess the effectiveness of the *judge* agent, we invite 5 well-trained annotators to assess the final inquires approved by the *judge* agent, evaluating the extent to which they align with the intended dimensions. The pass rates for each dimension on 500 randomly selected problems are shown as follows (Since Dimension 2 relies on rule-based sentence disruption, there is no need for a *judge* agent):
>
> Dimension |D1|D3|D4|D5|D6|D7|D8|D9
>  -|-|-|-|-|-|-|-|-
> Pass rate (*judge*)|0.984|0.992|0.964|0.986|0.986|0.952|0.990|0.950
>
> For the *reference* agent, the correct answers for Dimensions 1 to 4 are simply the answers to the original problems, so their corresponding *reference* agents do not rely on LLM generation and no additional evaluation are required. For Dimension 9, as explained in lines 346-347, the *reference* agent can directly use the value masked by the *inquiry* agent as the answer. For Dimensions 5 to 8, we also invite 5 annotators to assess the answers (i.e., outputs of *reference* agents) for 500 problems. From the table below, the accuracy for each dimension is found to exceed 95%.
>
> Dimension |D5|D6|D7|D8
>  -|-|-|-|-
> Pass rate (*reference*)|0.954|0.968|0.952|0.986
>
> These results demonstrate that our agents are indeed capable to produce high-quality results. Based on your comments, we will supplement more human evaluation results in the revised paper.
>
> $\bf{Q5}$:This work also collects MExam. However, I know nothing about it from the contents.
>
> $\bf{A5}$: Thank you for pointing that out. As mentioned in Section 4.1, our MExam dataset consists of 6,353 questions that were manually collected from real exams. These questions come from actual Chinese exams presented in 50 exercise books. Unlike MATH and GSM8K, MExam covers the full K-12 mathematics curriculum, allowing us to perform a more comprehensive evaluation. This broader coverage not only provides valuable insights for assessing LLM performance across a wide range of mathematical topics but also helps mitigate potential data leakage issues. Based on your concern, we will include additional details about the dataset construction process to provide more clarity.
>
> We sincerely hope our rebuttal can address your concerns.

---

> > ### Author Response · Authors · 2024-11-27
> >
> > We wish to once again express our great appreciation for the time you have taken to review our paper. We would appreciate your feedback on whether your main concerns have been adequately addressed. We truly value your understanding and support, and will carefully revise the paper according to your suggestions. Thank you very much!

---

> ### Comment · Area_Chair_x4XQ · 2024-11-27
>
> Dear Reviewer WAEw,
>
> Thank you for your efforts reviewing this paper. Can you please check the authors' response and see if your concerns have been addressed? Thank you!

---

### Note · Authors · 2024-12-15

**Comment:**

We would like to thank the Area Chair for their efforts in prompting the reviewers and facilitating the review process. Unfortunately, we regret that our rebuttal did not receive a positive response from the majority of the reviewers. We also believe that some of the concerns raised do not accurately reflect the contributions of our work. Therefore, we have decided to withdraw our submission from the ICLR review process.

**Withdrawal Confirmation:**

I have read and agree with the venue's withdrawal policy on behalf of myself and my co-authors.